# Investigating Racial Differences in Allostatic Load by Educational Attainment among Non-Hispanic Black and White Men

**DOI:** 10.3390/ijerph19095486

**Published:** 2022-04-30

**Authors:** Danielle R. Gilmore, Tzitzi Morán Carreño, Hossein Zare, Justin X. Moore, Charles R. Rogers, Ellen Brooks, Ethan Petersen, Carson Kennedy, Roland J. Thorpe

**Affiliations:** 1Trachtenberg School of Public Policy and Administration, George Washington University, Washington, DC 20052, USA; 2Graduate School of Education and Human Leadership, George Washington University, Washington, DC 20052, USA; tzitzim@gwu.edu; 3Johns Hopkins Center for Health Disparities Solutions, Bloomberg School of Public Health, Johns Hopkins University, Baltimore, MD 21218, USA; hzare1@jhu.edu; 4Institute of Public and Preventive Medicine, Medical College of Georgia, Augusta University, Augusta, GA 30912, USA; jusmoore@augusta.edu; 5Department of Family & Preventive Medicine, University of Utah School of Medicine, Salt Lake City, UT 84132, USA; charles.rogers@utah.edu (C.R.R.); ellen.brooks@utah.edu (E.B.); ethan.petersen@hsc.utah.edu (E.P.); u1068835@utah.edu (C.K.); 6Program for Research on Men’s Health, Johns Hopkins University, Baltimore, MD 21218, USA; rthorpe@jhu.edu

**Keywords:** educational status, epidemiologic methods, health disparities, men’s health, allostatic load

## Abstract

Education continues to be a key factor contributing to increased access to critical life-improving opportunities and has been found to be protective against Allostatic Load (AL). The purpose of this study was to assess AL among Non-Hispanic (NH) White and NH Black men with the same level of education. We used 1999–2016 National Health and Nutrition Examination Surveys (NHANES) data with an analytical sample of 6472 men (1842 NH Black and 4630 NH White), and nine biomarkers to measure AL, controlling for various demographic and health-related factors. NH Black men had a higher AL score than NH White men (39.1%, 842 vs. 37.7%, 1,975). Racial disparities in AL between NH Black and NH White men who have a college degree or above (PR: 1.49, CI: [1.24–1.80]) were observed. Models posited similar AL differences at every other level of education, although these were not statistically significant. The findings reveal that socioeconomic returns to education and the societal protective mechanisms associated with education vary greatly between White and Black men.

## 1. Introduction

The relationship between education and health has been widely studied, leading to a persistent and robust association between the two [1,2,3,4]. At least four conceptualizations explain why education leads to better health outcomes [1,5,6,7,8,9,10,11,12] The first argues that individuals with more years of education have greater health literacy and may be more mindful of their overall health [10,13]. The second suggests that more educated individuals have greater access to health resources, including high-quality medical care [9,13]. The third concept points to the association between low education and low socioeconomic status (SES). This combination has specific biological pathways, increasing the risk of chronic illness [3,8,9,10]. Finally, a fourth consideration involves the relationship between poor health and lower educational attainment (reverse causality) [7].

One specific health indicator that has been correlated with education is Allostatic Load (AL). Individuals with lower educational levels are thought to experience increased environmental, psychological, and behavioral challenges associated with low SES, the effects of which accumulate over an individual’s lifetime and place added stress on the body. This is called Allostatic Load or AL. Many studies have observed this negative relationship between education and AL, suggesting that higher educational attainment has a protective effect on AL and a series of other adverse conditions, such as decreased cognitive and physical function, heart disease, stroke, diabetes, and overall increased mortality [1,9,14,15]. 

While a large body of research has shown significantly higher AL among Blacks than Whites [1,6,8,12,16,17,18,19,20,21], previous studies of AL have rarely analyzed the interaction between education and race/ethnicity explicitly. Examining such an interaction is important because research has shown differential returns to social capital based on race/ethnicity. Racial and ethnic minorities are more likely to live in neighborhoods with fewer resources, attend lower-quality schools, be uninsured, and display other repercussions of the relationship between race/ethnicity and poverty that have lasting impacts [10,22,23,24,25,26]. These disparities suggest that the observed differences result from racial minorities having limited access to health-protective socioeconomic factors compared to whites. Moreover, the weathering hypothesis suggests that experiences associated with racism bring social stress that, in turn, contributes to the exacerbation of poor health outcomes in populations of color [8,9,10,27,28].

The few studies that have examined the impact of the interaction between education and race on AL confirm the disproportional burden of AL on minorities of color [3,10,17] However, they also reveal that racial differences are larger for those with higher levels of education [3,10]. These findings suggest that although educational attainment is a good predictor of AL, its protective effect is not the same for all members of a population group. The results also point to the influence of stressors such as systemic racism and self-perceived discrimination on the Black population’s health.

The literature presents several limitations. The samples of certain studies are proportionally small or focus exclusively on women, which restricts the extent to which their findings can be generalized. In some cases, the period of study covers only a couple of years or is dated. This study aimed to extend previous research by focusing on the health experiences of Black men throughout a comprehensive period (1999–2016) by assessing AL in non-Hispanic White and non-Hispanic Black men with the same level of education. We hypothesized that the association between race and AL would be elevated in NH Black men compared to NH White men regardless of their educational attainment.

## 2. Materials and Methods

### 2.1. Sample

Data were obtained from the National Health and Nutrition Examination Surveys (NHANES). This study’s original sample consisted of 92,062 participants for nine waves of the NHANES data between 1999 and 2016. NHANES is a cross-sectional in-person survey of adults 18 years of age or older [29]. Since 1999, NHANES has been utilizing survey data to identify the health, functional, and nutritional status of the US population, and the interrelationships of these. NHANES public-use data files are released biannually (e.g., NHANES 1999–2000, NHANES 2001–2002, NHANES 2003–2004) [30]. The cross-sectional surveys create a nationally representative sample of the civilian noninstitutionalized population. NHANES oversamples low-income individuals: participants aged between 12 and 19 years and adults aged 60 years and older, African Americans, and Mexican Americans [31]. These surveys use a stratified, multistage probability sampling design [31]. NHANES conducts a two-part data collection. First, participants provide their health history, health behaviors, and risk factors via at-home interviews. Second, participants receive a detailed physical examination at a mobile examination center [29,30,32,33] In this analysis, we used the combined 1999–2018 NHANES surveys and included all men who: (1) had valid data on the nine biomarkers used to create the AL composite variable, (2) self-identified as NH White or NH Black, and (3) were at least 20 years old. We yielded an analytic sample of 6472 men, including 1842 NH Black and 4630 NH White men.

### 2.2. Outcome Variable: Allostatic Load

The outcome variable for this study was AL. The AL metric entails the significant system dysregulation and cumulative burden on various organ systems/pathways (e.g., immune, cardiovascular, and metabolic) stemming from the frequent exposure to persistent physiological responses to stressful environments [6,9,12,14]. For this study, we used the summation of biomarker factors into one accumulative index to capture the cumulative toll of these risk factors, which provides a more accurate prediction of health outcomes than individual risk factors or other combinations of such factors [34]. Examining only NH Black and NH White men, we computed AL based on nine biomarkers, including BP systolic (mm Hg), BP diastolic (mm Hg), pulse rate (beats/min), glycohemoglobin (%), direct HDL-cholesterol (mg/dL), total cholesterol (mg/dL), serum Albumin (g/dL), body mass index (BMI), and estimated Glomerular Filtration Rate (eGRF). In detail, we calculated AL by first identifying each biomarker as high-risk (1) or not (0). Values were considered high risk above the 75th percentile, except in HDL-cholesterol, where high risk was defined as below the 25th percentile. After scores were determined based on risk, each biomarker’s scores were summed to obtain the AL. Finally, we created a dummy variable to identify individuals with a high AL (1) vs. low AL (0) (1, if AL ≥ 4; 0, if AL < 4) [8,35].

### 2.3. Main Independent Variable

The primary independent variables were race/ethnicity and education. Men reported their highest level of education, which was then categorized as less than high school; high school diploma or general equivalency diploma (GED); some college or associate degree; or bachelors’ degree and beyond. Men also reported their race as White or Black/African American and their ethnicity as Hispanic or not Hispanic. A variable was created to identify the racial/ethnic groups: NH White men (“White men” hereafter) and NH Black men (“Black men” hereafter).

### 2.4. Covariates

Demographic variables included age (years), married (1 = yes, 0 = no), income level (<USD 34,999, USD 35,000–74,999, ≥USD 75,000), race/ethnicity (1 = White, 0 = Black), and education level, with four categories (less than high school graduate; high school graduate or general equivalency diploma equivalent recipient; some college or associate degree; college graduate or above). Health-related characteristics included having health insurance (1 = yes, 0 = no), smoking and drinking status (0 = never, 1 = current, 2 = former), self-reported health (1 = fair/poor health, 0 = good/excellent health), and physical inactivity (1 = physically inactive or sedentary, 0 = physically active). We controlled for demographic and health-related characteristics that are associated with race/ethnicity in previous literature [2,3,36]. 

### 2.5. Analytic Strategy

The mean and proportional differences between White and Black men for high AL, demographic, SES, and health-related characteristics were evaluated using Student’s *t*-tests and chi-square tests. Since obesity was greater than 10% in this sample, a modified Poisson regression was used with a dichotomous AL variable [37,38]. Two models were tested to examine the relationship between race (independent variable) and high AL (dependent variable) while controlling for possible confounders. In model 1, we examined the association between race and AL by adjusting for all covariates. In model 2, we stratified our analysis by education and similarly adjusted for all other covariates. All analyses were weighted using the NHANES individual-level sampling weights for 1999–2016 [32]. We considered all *p*-values < 0.05 statistically significant, and all tests were two sided. All statistical procedures were performed using STATA statistical software, version 15.

## 3. Results

The distributions of characteristics by race were compared (see Table 1). Black men had a higher prevalence of AL when compared to White men (39.1% (n = 842, *p*-value: <0.001) vs. 37.7% (n = 1975, *p*-value: <0.001)). Black men were more likely to drop out of high school than White men (26.1% (n = 538, *p*-value: <0.001) vs. 12.5% (n = 831, *p*-value: <0.001)). Black men were less likely to be married than White men (52.9% (n = 1010, *p*-value: <0.001) vs. 71.5% (n = 3,211, *p*-value: <0.001)). Compared to White men, Black men were more likely to earn under USD 35,000 annually (40.4% (n = 764, *p*-value: <0.001) vs. 23.5% (n = 1657, *p*-value: <0.001)) and less likely to have current insurance coverage (70.1% (n = 1364, *p*-value: <0.001) vs. 86.1% (n = 3,924, *p*-value: <0.001)). Black men were more likely to be current smokers (32.7% (n = 590, *p*-value: <0.001) vs. 23.7% (n = 1106, *p*-value: <0.001)) but less likely to be current drinkers than White men (79.1% (n = 1375, *p*-value: <0.001) vs. 86.3% (n = 3,778, *p*-value: <0.001)). Black men were more likely to report fair or poor health (20.8% (n = 434, *p*-value: <0.001) vs. 14.6% (n = 880, *p*-value: <0.001)), more likely to be non-physically active (42.0% (n = 854 *p*-value: <0.001) vs. 37.1% (n = 1963, *p*-value: <0.001)) and less likely to currently be employed when compared to White men (69.5% (n = 1042, *p*-value: <0.001) vs. 72.9% (n = 2608, *p*-value: <0.001)).

The interaction between education and race was statistically significant (*p*-value: <0.001). We then stratified our analysis by education. The prevalence risk ratios obtained from the modified Poisson regression examining the relationship between AL and all covariates are shown in Table 2. Black men were 14% more likely to have higher AL with a prevalence risk (PR) of 1.14 and a confidence interval (CI) of (1.05–1.25). Compared to men who did not complete high school, men with a high school diploma or GED had a PR of 1.13 and a CI of (1.00–1.28). Men with some college or associate degrees were 17% more likely to have a higher AL (PR: 1.17, CI: (1.03–1.33)). College education appears to serve as a protective mechanism against AL. Men with a college degree or above were five percent less likely to report AL factors than men who did not complete high school (PR: 0.95, CI: (0.82–1.44)).

The association between AL and race by educational level is shown in Table 3. Compared to White men, only Black men who earned a college degree or above experienced the statistically significant racial disparities in AL (PR: 1.49 and CI: 1.24–1.80). Black men who earned a college degree or above have an AL 1.49 times higher than equally educated White men. Across other categories, a similar adjusted prevalence is present, although this is not statistically significant. 

## 4. Discussion

Our study aimed to assess AL among White and Black men with the same level of education. When considering the association between education and AL in Black and White men, our results refute previous trends in the literature, which conclude that Black men have a higher prevalence of AL than White men with the same level of education. Research has demonstrated that members of underrepresented populations experience greater stress than their White peers, even when controlling for factors such as discrimination or SES [8,12]. Nevertheless, our study contributes to the literature via our key finding that Black male college graduates have a higher AL than their White male counterparts. Our results indicate that education alone is not protective against the negative health impacts of chronic stress. Moreover, this prevalence association was present for every level of education, but there was only a statistically significant difference between Black and White men with a bachelor’s degree or higher.

Given that the AL burden partially explains the higher mortality among Black men juxtaposed to their racial and ethnic counterparts, further investigating the interplay between health disparities and AL is essential [1,3,6,9,14]. Despite declines in mortality coupled with increases in life expectancy, the literature overwhelmingly portrays racial disparities between NH Black and NH White men [4,6,8,11,12,13,39,40,41]. Race-based health disparities are often attributed to lower SES consequences, including education [13]. However, research shows that even with less education, White men significantly outlive Black men [11]. Additionally, a study by Borrell et al. (2010) revealed that Blacks’ AL is significantly higher than their white peers; Blacks’ AL was more aligned with Whites ten years older, even after controlling for income [1]. Thus, SES may not be the primary driver of poor health outcomes. 

## 5. Conclusions

Our findings are consistent with the prior research examining the link between education and AL, which have a negative correlation, with more educated individuals often have a lower AL [4,6,8,12,13,39,40,41]. For instance, Geronimus et al. (2006) found that the AL mean scores for Blacks were similar to those of Whites who were ten years older [8]. This study implies that race/ethnicity may play a more significant role in AL than previously thought, especially in its impact on Black men. Even after controlling for education and other SES-related factors, our results confirm racial differences in AL. Racial disparities are fueled by various manifestations, including explicit discrimination in the form of racism, income inequality, or other external factors associated with minority status [1,2,3,6,10,11,12,42]. Additional research is needed to examine how racism drives the increased risk of high allostatic load, even when controlling for education.

Our paper contributes to the substantive and methodological literature on men’s health research. Our study’s two key strengths include using a nationally representative sample of U.S. Black and White men (age ≥ 20) to conduct the analysis. We also used nine NHANES surveys data between 1999 and 2016, which resulted in a sufficient number of Black and White men from a large and diverse dataset to create a subpopulation for analysis. Despite these strengths, our paper has several limitations. First, since NHANES is a cross-sectional survey, neither causality nor temporality can be established. Second, another concern was that NHANES data are self-reported and may be subject to bias. Finally, we could not subdivide education into additional categories (e.g., bachelors, masters, etc.) due to the small sample size of Black men with post-secondary and graduate degrees. Despite these limitations, to the authors’ knowledge, this is the first study to examine AL between Black and White men within the same levels of education exclusively. Conversely, the inclusion of covariates permitted our team’s analyses to account for confounding factors (e.g., health and income), which may impact AL prevalence. 

We found that Black men with a college degree had higher AL than equally educated White men. Accordingly, these findings revealed that socioeconomic returns to education and the societal protective mechanisms associated with education varied greatly between White and Black men. Education does not protect minorities from discrimination and the resulting chronic stress. Future research should investigate how stereotype threat and racism—racial microaggressions included—may drive disparities in AL despite similar SES and educational attainment. A further subgroup analysis may provide insight into how AL differently impacts Black men for other chronic diseases, even after controlling for SES and education [1,2,3,10,12,36].

## Figures and Tables

**Table 1 ijerph-19-05486-t001:** Distribution of select characteristics of non-Hispanic Black and non-Hispanic White men using 1999–2016 NHANES.

	Non-Hispanic Black (n = 1842)	Non-Hispanic White (n = 4630)	*p*-Value
**Age (mean ± SE)**	43.1 ± 0.4	47.8 ± 0.3	<0.001
**Level of Education (%)**			
**Less than High School**	26.1 (538)	12.5 (831)	<0.001
**High School Diploma or GED**	28.4 (524)	24.9 (1187)	<0.001
**Some College or Associate Degree**	31.1 (528)	30.5 (1315)	<0.001
**College Degree and Above**	14.3 (248)	32.1 (1293)	<0.001
**Marital Status (%)**	52.9 (1010)	71.5 (3211)	<0.001
**Income**			
**USD 0–USD 34,999**	40.4 (764)	23.5 (1657)	<0.001
**USD 35,000–USD 75,000**	38.5 (677)	37.3 (1589)	<0.001
**USD 75,000 or Above**	18.8 (347)	38.3 (1331)	<0.001
**Missing**	2.4 (54)	0.9 (53)	<0.001
**Health Insurance Coverage (%)**	70.1 (1364)	86.1 (3924)	<0.001
**Smoking Status**			
**Never**	50.2 (846)	43.7 (1823)	<0.001
**Former**	17.1 (401)	32.5 (1699)	<0.001
**Current**	32.7 (590)	23.7 (1106)	<0.001
**Drinking Status**			
**Never**	10.8 (173)	7.1 (347)	<0.001
**Former**	10.0 (191)	6.6 (348)	<0.001
**Current**	79.1 (1375)	86.3 (3778)	<0.001
**Self-Reported Fair/Poor Health (%)**	20.8 (434)	14.6 (880)	<0.001
**Not Physically Active (%)**	42.0 (854)	37.1 (1963)	0.002
**Employed**	69.5 (1042)	72.9 (2608)	0.014
**Allostatic Load (if AL ≥ 4) (%)**	39.1 (842)	37.7 (1975)	<0.001

**Table 2 ijerph-19-05486-t002:** Association between race and Allostatic Load among Non-Hispanic Black and Non-Hispanic White Men in the 1999–2016 NHANES.

	Prevalence Ratio (PR)	Confidence Interval
**Non-Hispanic African American ***	1.14	1.05–1.25
**Age**	1.02	1.01–1.02
**Married**	1.04	0.94–1.16
**Level of Education**		
**Less than High School**	-	-
**High School Diploma or GED**	1.13	1.0–1.28
**Some College or Associate Degree**	1.17	1.03–1.33
**College Degree and Above**	0.95	0.82–1.11
**Income**		
**USD 0–USD 34,999**	-	-
**USD 35,000–USD 75,000**	0.99	0.82–1.19
**USD 75,000 or Above**	1.01	0.75–1.35
**Health Insurance Coverage**	1.11	0.85–1.44
**Smoking Status**		
**Never**	-	-
**Former**	0.88	0.69–1.13
**Current**	0.72	0.55–0.960
**Drinking Status**		
**Never**	-	-
**Former**	1.31	0.87–1.97
**Current**	1.39	1.89
**Self-Reported Fair/Poor Health**	1.10	1.33
**Physically Inactive**	1.46	1.80
**Employed**	1.02	1.45

***** Non-Hispanic White men were the reference group.

**Table 3 ijerph-19-05486-t003:** Association between race and Allostatic Load by education level in Non-Hispanic Black and Non-Hispanic White men in the 1999–2016 NHANES Files.

	Did Not Complete High School	High School Diploma or GED	Some College or Associate Degree	College Degree or Above
**Non-Hispanic Black ***	1.04 (0.87–1.25)	1.11 (0.96–1.29)	1.08 (0.93–1.26)	1.49 (1.24–1.80)

***** Non-Hispanic White men were the reference group.

## Data Availability

The National Health and Nutrition Examination Survey (NHANES) is a public use dataset available at https://www.cdc.gov/nchs/nhanes/index.htm, accessed on 20 April 2021.

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
