# Peer review of "Investigating Racial Differences in Allostatic Load by Educational Attainment among Non-Hispanic Black and White Men"

_ijerph, 2022, doi:10.3390/ijerph19095486_

Round 1
Reviewer 1 Report
Review report is attached.

Author Response
Point 1: In line 21, define “AL” the first time it is referenced.
Response 1: We thank the reviewers for the suggestion and have updated the abstract accordingly.
Point 2: Line 26 states, the authors state, “Among Racial disparities in…” Remove “Among.”
Response 2: We thank the reviewers for this comment and have updated the language accordingly
Introduction
Point 3: While the introduction presents interesting information, the authors highlight several limitations/gaps in the current literature base that are not addressed by their paper so it’s not clear why the information is presented. For example: a. In lines 61-62, the authors state, “However, existing literature has not elucidated whether a causal link exists between education and the biomarkers considered for AL indices.” Is this an aim of the paper? If so, it needs to be explicitly called out in the aims statement at the end of the introduction. b. In lines 81-84, the authors state, “Moreover, few studies have examined AL among Black men (Baciu et al., 2017; Rog- 81ers et al., 2020). While those studies have helped, they fall short on Black mens’ experiences. Many unanswered questions remain regarding the extent to which education contributes to racial disparities among NH White and NH Black men.” What do you mean by current studies fall short on Black men’s “experiences” and how does this paper address that in a way that previous studies have not?
Response 3: We thank the reviewers for this comment and have updated the introduction to better align the literature reviewed with the aims of the paper.
Point 4: In lines 86-87, the authors state, “We hypothesized that the association between race and AL would be most significant among Black men regardless of their educational attainment.” Two concerns:
- First, the hypothesis does not align with the gaps/limitations and literature presented (this can be addressed by addressing the comment I made above (comment #3) and below (comment #5)).
Response 4a: We thank the reviewers for this comment and have updated the introduction to better align the literature reviewed with the aims of the paper.
- Two, I suggest removing the language “most significant.” It would be more appropriate to state, “We hypothesized that the association between race and AL would be greater among Black men than White men, regardless of educational attainment.” “Most significant” has the connotation that you are comparing the strength of the significance (i.e., p-values), and I expect statisticians would push back on the way in which the hypothesis is currently framed.
Response 4b: We thank the reviewers for this comment and have edited the language accordingly.
Point 5: Overall, this introduction does not appropriately align with the stated aim of the paper. The way it is introduced leads the reader to believe the paper will be focused on establishing the relationship between education/SES and AL. However, that's not the stated aim of the paper and it does not align with most of the results presented. Given that the aim is to “assess AL in NH White and NH Black men with the same level of education,” and that the authors hypothesize that “the association between race and AL would be most significant among Black men regardless of their educational attainment,” the authors should provide us with the rationale for why we should expect these findings. What gaps exist in the current literature to influence your stated aim? What literature exists to support your hypothesis? There is an overemphasis on education/SES influencing AL in the current introduction. While it's important to acknowledge this literature, it should not be the overall focus based on the stated aim of the paper. For example, I would expect the authors to touch on the faulty assumption that racial differences in AL are primarily explained by differences in education and SES between Black and White men. The introduction should set the reader up to question this logic.
Response 5: We thank the reviewers for this comment and have updated the introduction to better align the literature reviewed with the aims of the paper.
Materials and Methods
Point 6: In the Outcome Variable: Allostatic Load section, the authors should indicate whether the AL index described is a validated measure of AL or if the index was validated in the current sample. Report reliability statistics.
Response 6: For this paper we have calculated AL using used measure Bey et al. 2018 and Geronimus et al., 2006 approaches. We included nine biomarkers including:
BP systolic (mm Hg),
BP diastolic (mm Hg),
pulse rate (beats /min),
glycohemoglobin (%),
direct HDL-cholesterol (mg/dL),
total cholesterol (mg/dL),
serum Albumin (g/dL),
body mass index (BMI),
estimated Glomerular Filtration Rate (eGRF).
For this analysis we used the allostatic load score computed as sum of all 9 biomarkers, existence of a condition considered as 1 and otherwise 0. Values above the 75th percentile were defined as high risk for all the biomarkers, with the exception of HDL and serum albumin, for which values below the 25th percentile were defined as high risk. We then created a dummy variable if ALS>=4 as 1 otherwise zero and used that variable for the Poisson model. High AL was based on those men who had 4 or more biomarker considered to be high risk.
The scale is more likely a clinical scale and broadly used by many studies, so we did not compute any reliability test.
Point 7: In the Covariates section, the authors classify race/ethnicity and education as covariates when they have already been classified as the primary independent variables. They shouldn’t be categorized as both unless there are truly instances in which they are used as both. This does not appear to be the case.
Response 7: Thank you for noticing this we have removed the race/ethnicity and education from co-variates section.
Point 8: In the Covariates section, drinking status is operationalized in the same way as smoking status which is not how we typically see drinking status operationalized in the literature. I recommend re-coding the categories for this variable using the categories typically used in the literature (e.g., never, occasional/moderate, heavy drinker). Otherwise, use of these categorizations should be reported as a limitation unless you can provide support (ideally from the literature) for your decision to use them.
Response 8: We thank you reviewer, for this suggestion, but the NAHNES data has some limitations for using the suggested variables; for example, if we wanted to use them (never, occasional/moderate, heavy drinker) definition, we have to lose more than 50% of the sample because of the missing observation, so we treated this variable by using two variables: 1) Do you now smoke cigarettes with three categories (every day, some days and not all) and 2) Smoked at least 100 cigarettes in life (yes/no). We then created a new variable:
- Never smoked if never smoked at least 100 cigarettes
- Former smoker if smoked at least 100 cigarettes but answer “not all” to the 2nd question.
- Current smokers if smoke every day or some days
Our measure has been used widely. Here are a few examples:
Gilmore, D. R., Whitfield, K. E., & Thorpe Jr, R. J. (2019). Is there a difference in all-cause mortality between non-Hispanic Black and non-Hispanic White men with the same level of education? Analyses using the 2000–2011 National Health Interview Surveys. American journal of men's health, 13(1), 1557988319827793.
Duru, O. K., Harawa, N. T., Kermah, D., & Norris, K. C. (2012). Allostatic load burden and racial disparities in mortality. Journal of the National Medical Association, 104(1-2), 89-95.
Zare, H., Gaskin, D. D., & Thorpe, R. J. (2021). Income Inequality and Obesity among US Adults 1999–2016: Does Sex Matter? International journal of environmental research and public health, 18(13), 7079.
Point 9: In lines 151-153 of the Analytic Strategy section, the authors state, “Since obesity was greater than 10% in this sample, a modified Poisson regression was used…” It’s not clear why obesity influenced this decision, especially given that obesity is not mentioned anywhere else in the manuscript, is not mentioned as a factor influencing the relations being examined, and is not described as a variable measured in the study. The authors should clarify this.
Response 9: Thank you for noticing this, we have edited the text to: Since AL’s dummy variable was greater than 10% in this sample, a modified Poisson regression was used with a dichotomous AL variable.
Results
Point 10: In lines 182-88, the authors interpret the relationship between education and AL across the full sample. I think it would be beneficial to see this information stratified by race as well, especially given that the authors hypothesize that race is associated with AL regardless of education. The full table 2 could be stratified by race. This could provide additional context for the results seen in table 3 (primary aim). Also, given the authors’ emphasis on these results (nearly an entire paragraph), it seems like the relation between education and AL should be an additional aim or sub-aim.
Response 10: We thank the reviewer for pointing this out. However, in our analyses we conceptualize education as the moderator. Therefore, we stratified our analyses on level of education. We recognize that race is an important social construct that is often time strongly linked to education. We also believe that this would be a different research question/objective to achieve.
Point 11: In lines 193-194, the authors state, “Across other categories, similar adjust prevalence present…” It’s not clear what “similar adjust prevalence” means. I suspect there’s a typo here. The authors should clarify.
Response 11: We thank the reviewer for pointing this out. We have now revised our text.
Point 12: In Table 3, the authors should note the control variables in a footnote. Tables should be standalone in that the reader should not have to refer to the text or another table to interpret it.
Response 12: We have now added the controls variables as a footnote.
Discussion
Point 13: In lines 217-218, the authors state, “Given that many of the AL variables show trends that differ by race and ethnicity, identifying the role of racial disparities on AL is essential (Borrell et al., 2010; Ding et al., 2019; Duru et al., 2012; Hamdi et al., 2016; Rogers et al., 2020). It’s not clear what is meant by this statement. What role do disparities have on AL? There can be disparities in AL and disparities in the indictors of AL, but it sounds like the authors are saying disparities impact AL. What disparities are you referring to here and in what way? Maybe the authors mean that in general inequities in health can influence AL among Black men and these are the inequities we need to identify. Or maybe the authors intended to say identifying the role of AL on disparities in health is essential. I think the vagueness of the statement adds to the ambiguity. It would be helpful if the authors clarified what they mean here.
Response 13: Thanks for bringing this to our attention. Accordingly, we have updated the text to read as follows for this section: “Given that AL burden partially explains higher mortality among Black men juxtaposed to their racial and ethnic counterparts, further investigating the interplay between health disparities and AL is essential (Borrell et al., 2010; Ding et al., 2019; Duru et al., 2012; Hamdi et al., 2016; Rogers et al., 2022).”
Point 14: In lines, 217-230, the authors seem to be making the link between AL and the documented disparities in health between White and Black men. However, this paragraph comes across as a series of disjointed facts about disparities and AL. It would help to make the connection between the various statements in this paragraph clearer, more cohesive, and a logical flow from one statement to the next (how one leads to/is connected to the other). Maybe a couple of transition sentences would help.
Response 14: Thanks for bringing this to our attention. Accordingly, we have updated the opening paragraph of our discussion section extensively to increase its clarity. Specifically, it now reads: ‘The purpose of this study was to assess AL among White and Black men with the same level of education. When considering the association between education and AL in Black and White men, our results refute previous trends in the literature which conclude that Black men have a higher prevalence of AL than White men with the same level of education. Research has demonstrated that members of underrepresented populations experience greater stress than their White peers, even when controlling for factors like discrimination or SES (Geronimus et al., 2006; Tomfohr et al., 2016). Yet, our study contributes to literature via our key finding that Black male college graduates have a higher AL than their White male counterparts. This indicates that education alone is not protective against the negative health impacts of chronic stress. Moreover, this prevalence association was present for every level of education, but there was only a statistically significant difference between Black and White men with a bachelor’s degree or higher.’
Point 15: In lines 231-34, the authors state that “Our findings are consistent with the prior research examining the link between education and AL found a negative correlation, with more educated individuals often having a lower AL...” However, this is not quite true. Those with more education tended to have high AL, except for the college category which was not significant. Also, a PR of 0.95 with a wide confidence interval is arguably more of a lack of an association than a negative association. I recommend modifying this statement.
Response 15: Duly noted. We have updated the sentence to read: ‘When considering the association between education and AL in Black and White men, our results refute previous trends in the literature which conclude that Black men have a higher prevalence of AL than White men with the same level of education.’
Point 16: In line 240, the authors state, “Racial disparities are fueled by various manifestations, including explicit discrimination…” Manifestations of what? Do you mean manifestations of racism? For example, discrimination would be a manifestation of racism. Please clarify this language.
Response 16:Thank you for your comment. The manifestations we are referring to immediately follow: “including explicit discrimination in the form of racism, income inequality, or other external factors associated with minority status.”
Point 17: In lines 244-45, the authors state, “Additional research is needed to examine how racism drives increased risk of high allostatic load, even when controlling for education.” This is the first time racism is mentioned in the discussion as being related in any way to allostatic load. I agree this is an important place of future research but a connection between racism and your findings/allostatic should be made prior to suggesting more research is needed. Otherwise, the point will appear to come out of nowhere for readers.
Response 17: Lines 240-244: “Racial disparities are fueled by various manifestations, including explicit discrimination in the form of racism, income inequality, or other external factors associated with minority status (Borrell et al., 2010; Duru et al., 2012; Gilmore et al., 2019; Howard & Sparks, 2015; Robert A Hummer & Elaine M Hernandez, 2013; Moe & Gilmour, 1995; Rogers et al., 2020; Tomfohr et al., 2016).
Point 18: In lines 255-56, the authors state, “Finally, we could not subdivide education into additional categories (e.g., bachelors, masters) due to small sample size of Black men with post-secondary and graduate degrees.” This should appear a couple of lines before its current placement, with the other limitations mentioned. The authors seemingly conclude the limitations in line 253, but then add in this one a couple lines later.
Response 18:Thank you for your comment, this sentence was moved to lines 253-255.
Point 19: In line 264-65, the authors state, “Future research should investigate the effect that racism has on AL within more finite education categories.” It’s not clear what is meant here. Do the authors mean something more along these lines: “investigate how racism may drive disparities seen despite similar SES/education”?
Response 19: Thank you. The sentence was edited to the following: “Future research should investigate how racism may drive disparities seen in AL despite similar SES and educational attainment.”
Point 20: Overall, the discussion could be strengthened. Several of the points made came across as vague and too general. So it’s not clear why the authors are making certain statements. The authors are headed in the right direction, and I believe I can infer their intentions, but this section would be much stronger if they were more specific about the future work needed and why. Along those lines, the discussion section is also missing an adequate discussion of why the authors think they’re seeing the results they saw in their study. It appears that may have alluded to this a bit in lines 261-64 but it should be expanded and appear much earlier on in the discussion. This will also help strengthen the “why” for future research.
Response 20: Lines 213-215: “Our key findings indicate, among college graduates, Black men have a higher AL than and White, indicating that education alone is not protective against the negative health impacts of chronic stress (PR: 1.49 [CI: 1.24 – 1.80]).”

Reviewer 2 Report
The topic of this paper is important, but the manuscript is in desperate need of extensive copy-editing. I was surprised that the writing was so poor. Many of the sentences are almost incomprehensible because of missing words, grammatical errors, or other infelicities that seriously detract from the presentation.
The authors should more strongly justify their methodological decisions, such as the operational definition of the dependent variable, allostatic load (see lines 110-128).
The overall Black-White difference in AL -- 39.1% versus 37.7%, respectively (lines 162-164) does not seem that great. The difference may be statistically significant, yet I'm not sure that it is substantively significant.
The main finding appears to be that the Black-White difference in AL is significant only for men with a college degree or above (Table 3). This result is intriguing because highly-educated Blacks probably have, on the average, more interactions with Whites (in the workplace, for example) than do less-educated Blacks. Perhaps highly-educated Blacks find such interactions to be particularly stressful for numerous reasons, one of which may be "stereotype threat." Perhaps these interactions expose Blacks to more personal slights and "microaggression." The authors might explore these possibilities.
A potentially important explanatory variable that is missing (lines 137-147) is neighborhood context. Blacks at all socioeconomic and education levels tend to live in residential areas that have extraordinarily high crime rates compared to the neighborhoods of their White counterparts. Virtually all of this crime is Black-on-Black. Living under such conditions is, needless to say, highly stressful and contributes to Black-White differences in psychological trauma.
The paper's bibliography (lines 289-391) has numerous redundancies, with some references listed three and four times. Again, a professional copy-editor is needed to correct these mistakes.
Author Response
Reviewer Two
Dear Reviewer, we thank you for your most valuable comments and appreciate having had this wonderful opportunity to learn from you. We hope that our responses have addressed your comments effectively.
Point 1: The topic of this paper is important, but the manuscript is in desperate need of extensive copy-editing. I was surprised that the writing was so poor. Many of the sentences are almost incomprehensible because of missing words, grammatical errors, or other infelicities that seriously detract from the presentation.
Response 1 We thank the reviewer for this comment and have given the manuscript an additional read through to correct grammar, sentence structure and typos.
Point 2: The authors should more strongly justify their methodological decisions, such as the operational definition of the dependent variable, allostatic load (see lines 110-128).
Response 2
Point 3: The overall Black-White difference in AL -- 39.1% versus 37.7%, respectively (lines 162-164) does not seem that great. The difference may be statistically significant, yet I'm not sure that it is substantively significant.
Response 3: We agree with the reviewer; we have edited the text to address this comment:
Black men had a slightly higher prevalence of AL when compared to White men (39.1% vs. 37.7% [ n=1,975, p-value: 0.422]). However, it was not significant. We also edited the abstract.
Point 4: The main finding appears to be that the Black-White difference in AL is significant only for men with a college degree or above (Table 3). This result is intriguing because highly-educated Blacks probably have, on the average, more interactions with Whites (in the workplace, for example) than do less-educated Blacks. Perhaps highly-educated Blacks find such interactions to be particularly stressful for numerous reasons, one of which may be "stereotype threat." Perhaps these interactions expose Blacks to more personal slights and "microaggression." The authors might explore these possibilities.
Response 4: Great feedback. We have updated a sentence in our conclusion to read as follows: ‘Future research should investigate how stereotype threat and racism—racial microaggressions included—may drive disparities seen in AL despite similar SES and educational attainment.’
Point 5: A potentially important explanatory variable that is missing (lines 137-147) is neighborhood context. Blacks at all socioeconomic and education levels tend to live in residential areas that have extraordinarily high crime rates compared to the neighborhoods of their White counterparts. Virtually all of this crime is Black-on-Black. Living under such conditions is, needless to say, highly stressful and contributes to Black-White differences in psychological trauma.
Response 5 We thank the reviewer for the comment and have updated the text accordingly
Point 6: The paper's bibliography (lines 289-391) has numerous redundancies, with some references listed three and four times. Again, a professional copy-editor is needed to correct these mistakes.
Response 6 We thank the reviewers for this comment and have updated the bibliography,

Round 2
Reviewer 1 Report
I appreciate the authors' responses. Overall, they did a great job addressing my original concerns.
Reviewer 2 Report
None
This manuscript is a resubmission of an earlier submission. The following is a list of the peer review reports and author responses from that submission.